# Cytokines and Pancreatic Ductal Adenocarcinoma: Exploring Their Relationship with Molecular Subtypes and Prognosis

**DOI:** 10.3390/ijms25179368

**Published:** 2024-08-29

**Authors:** Laura Gutierrez-Sainz, Victoria Heredia-Soto, Ana Margarita Rodríguez-García, María Gema Crespo Sánchez, María Gemma Serrano-Olmedo, Marta Molero-Luis, Itsaso Losantos-García, Ismael Ghanem, Pablo Pérez-Wert, Ana Custodio, Marta Mendiola, Jaime Feliu

**Affiliations:** 1Medical Oncology Department, La Paz University Hospital, IdiPAZ, Paseo de la Castellana 261, 28046 Madrid, Spain; ismael.ghanem@salud.madrid.org (I.G.); juanpablo.perez@salud.madrid.org (P.P.-W.); ana.custodio@salud.madrid.org (A.C.); jaime.feliu@salud.madrid.org (J.F.); 2Translational Oncology Research Laboratory, Biomedical Research Institute, La Paz University Hospital, IdiPAZ, Paseo de la Castellana 261, 28046 Madrid, Spain; 3Centro de Investigación Biomédica en Red-Cáncer (CIBERONC), 28029 Madrid, Spain; marta.mendiola@salud.madrid.org; 4Pathology Department, La Paz University Hospital, IdiPAZ, Paseo de la Castellana 261, 28046 Madrid, Spain; anamargarita.rodriguez@salud.madrid.org; 5Clinical Analysis Department, La Paz University Hospital, IdiPAZ, Paseo de la Castellana 261, 28046 Madrid, Spain; mariagema.crespo@salud.madrid.org (M.G.C.S.); mariagemma.serrano@salud.madrid.org (M.G.S.-O.); marta.molero@salud.madrid.org (M.M.-L.); 6Biostatistics Department, La Paz University Hospital, IdiPAZ, Paseo de la Castellana 261, 28046 Madrid, Spain; itsaso.losantos@salud.madrid.org; 7Molecular Pathology and Therapeutic Targets Lab, Pathology Department, La Paz University Hospital, IdiPAZ, Paseo de la Castellana 261, 28046 Madrid, Spain; 8Cátedra UAM-AMGEN, Universidad Autónoma de Madrid, 28049 Madrid, Spain

**Keywords:** pancreatic ductal adenocarcinoma, cytokines, prognosis, molecular subtypes

## Abstract

Pancreatic ductal adenocarcinoma (PDAC) is characterized by its poor prognosis. The current challenge remains the absence of predictive biomarkers. Cytokines are crucial factors in the pathogenesis and prognosis of PDAC. Furthermore, there is growing interest in differentiating between molecular subtypes of PDAC. The aim of our study is to evaluate the association between the analyzed cytokines and the molecular subtypes of PDAC and to determine their prognostic value. Cytokine levels were measured in 73 patients, and molecular subtypes were analyzed in 34 of these patients. Transforming Growth Factor Beta 2 (TGF-β2) levels were independently associated with the basal-like and null subtypes. In patients with locally advanced and metastatic PDAC, elevated levels of interleukin (IL)-1α, IL-1β, IL-6, IL-8, IL-9, and IL-15 were associated with a higher risk of progression during first-line treatment, and increased levels of IL-1β, IL-6, IL-8, IL-9, and IL-15 were related to increased mortality. Furthermore, a significant association was observed between higher percentiles of IL-6 and IL-8 and shorter progression-free survival (PFS) during first-line treatment, and between higher percentiles of IL-8 and shorter overall survival (OS). In the multivariate analysis, only elevated levels of IL-8 were independently associated with a higher risk of progression during first-line treatment and mortality. In conclusion, the results of our study suggest that cytokine expression varies according to the molecular subtype of PDAC and that cytokines also play a relevant role in patient prognosis.

## 1. Introduction

Pancreatic ductal adenocarcinoma (PDAC) is the twelfth most common cancer and the sixth leading cause of cancer death worldwide [1]. In Spain, PDAC ranks seventh in terms of incidence and third in terms of causing death [2]. Despite advances in understanding potential risk factors and preventive measures, its incidence is expected to continue rising in the future and its prognosis remains poor [3].

At the time of diagnosis, 80–85% of patients with PDAC present with locally advanced or metastatic disease without the possibility of complete surgical resection, which is currently the only curative treatment [4]. In relation to the localization of metastases, the liver is the most common location, followed by the peritoneum and the lung, and less frequently, the bone [4]. In recent years, new first-line treatment options have been developed [5,6,7,8]. Between 2011 and 2013, FOLFIRINOX (oxaliplatin, irinotecan, leucovorin and fluorouracil) and the combination of gemcitabine and nab-paclitaxel demonstrated longer overall survival (OS) times compared to first-line gemcitabine monotherapy, with median OS times of 11.1 and 8.5 months, respectively [5,6]. In 2023, the NAPOLI-3 trial compared NALIRIFOX (oxaliplatin, liposomal irinotecan, leucovorin and fluorouracil) with the gemcitabine and nab-paclitaxel combination, showing longer OS (median OS of 11.1 and 9.2 months, respectively) [7]. More recently, the SEQUENCE study compared sequential treatment using gemcitabine and nab-paclitaxel, followed by FOLFOX (oxaliplatin, leucovorin and fluorouracil) administration, with the use of gemcitabine and nab-paclitaxel alone, revealing prolonged OS in the sequential arm, although it came with increased adverse events [8].

In recent years, increasing attention has been given to differentiating between molecular subtypes of PDAC [9,10,11,12,13,14,15,16,17,18,19,20,21]. On this point, distinct molecular classifications have been proposed, distinguishing between two to five molecular subgroups, with two of them commonly found among the proposed classifications: the so-called classical and basal-like subtypes [9,10,12,14,15,16,17,21]. GATA6 expression is correlated with the classical subtype and the expression of cytokeratin (CK) 5 and CK17 is associated with the basal-like subtype [17,18,19,20,21,22]. This differentiation is important since the basal-like subtype has a worse prognosis and different response to chemotherapy regimens, with better results achieved with gemcitabine-based treatments [9,10,16,17,23,24]. In addition, two other subtypes have been described, called mixed and null. The mixed subtype expresses both GATA6 and CK5/17, while the null subtype is characterized by the lack of expression of GATA6, CK5, and CK17 [21,22]. Therefore, the accurate identification of the molecular subtype could help in patient stratification and improve treatment strategies.

Furthermore, the lack of effective predictive and therapeutic biomarkers remains a major challenge in the management of this disease. In this context, the tumor microenvironment plays a crucial role in both the pathogenesis of this disease and its evasion of the immune system, contributing to progression and resistance to various treatment approaches [25,26]. Within the tumor microenvironment, the role of cytokines in the progression and development of PDAC has emerged as a promising area of research that deserves further exploration [27,28,29]. Cytokines have the potential to serve as valuable biomarkers for predicting cancer progression, treatment responses, and prognosis in patients with PDAC [30,31,32,33,34,35,36,37,38,39,40,41,42,43]. Specifically, interleukin (IL)-6, IL-8, and tumor necrosis factor alpha (TNF-α) are the cytokines with the most evidence in the literature related to poor prognosis in these patients [30,31,32,33,34,35,36]. IL-6 plays a multifaceted role in the development and progression of PDAC as it can act directly on tumor cells but also modulate the tumor microenvironment [35]. The primary function of IL-8 in PDAC is the promotion of angiogenesis, as it induces the production of VEGF [36]. TNF-α increases the invasiveness of PDAC cells and promotes tumor growth and metastasis development [37]. IL-10 and transforming growth factor beta (TGF-β) are two anti-inflammatory cytokines that appear to play a role in impairing the activity of cytotoxic T cells in the tumor microenvironment [30]. IL-10 also has potent antiangiogenic activity and can inhibit tumor growth and metastasis formation [38]. On the other hand, IL-10 can also indirectly inhibit the secretion of pro-inflammatory cytokines such as IL-6 and TNF-α [38]. TGF-β, for its part, promotes the formation of the tumor microenvironment and facilitates PDAC progression and dissemination [39]. Hu et al. [40] demonstrated that IL-9 significantly promoted the proliferation, invasion, and migration of PDAC cells; however, its effect on tumor cell apoptosis was insignificant. IL-17 also promotes carcinogenesis and tumor growth in PDAC [41,42]. The expression of IP-10 is a prognostic marker in PDAC patients and is associated with poorer survival [43]. IL-1α and IL-1β appear to play a complex and fundamental role in the host immune response to PDAC tumor cells [44].

The identification of specific cytokines, whose levels correlate with the prognosis of PDAC patients, could assist in selecting patients with differential prognoses and guide the selection of more effective therapies [34,45,46,47,48]. However, the possible relationship between molecular subtypes and cytokines has not been studied. Since the composition of the tumor stroma differs according to the molecular subtype [10], it is possible that there are also differences in cytokine levels.

The hypothesis of this study is that the simultaneous analysis of molecular subtypes of PDAC and the identification of specific cytokines may allow for the establishment of a correlation between certain cytokines and the molecular subtypes. Additionally, the detection of certain cytokines may also help to predict the prognosis of patients with PDAC and be related to the localization of metastases.

## 2. Results

### 2.1. Descriptive Analysis of Clinical and Analytical Variables

Initially, 210 patients with PDAC were selected. After thawing and processing plasma samples, 137 samples were determined to be unsuitable for cytokine analysis. Specifically, the main reasons for this were that in most patients (n = 112), plasma samples were insufficient to perform cytokine analysis, and on the other hand, in a few patients (n = 25) the samples did not retain the necessary integrity for rigorous cytokine analysis. Consequently, 73 patients were ultimately included in this study. Among these patients, cytological samples were available for 34 patients, enabling the analysis of the expression of GATA6 and CK17 via IHC (Appendix A).

Baseline clinical characteristics are summarized in Table 1. Overall, 33 patients (45%) were males, and 40 patients (55%) were females, with a median age of 65 years old (range: 37 to 86 years). The most common ECOG performance status was 0 or 1 (n = 24, 33% and n = 45, 62%, respectively), and only 4 patients (5%) had ECOG 2.

The pancreatic head was the most common location for the primary tumor (n = 41, 56%). Most patients were diagnosed with metastatic disease (n = 40, 55%), followed in frequency by locally advanced disease (n = 21, 29%), and 12 patients had localized/borderline disease (n = 12, 16%). In the 40 patients who presented with metastatic disease, the most common location of metastases was the liver (n = 33, 82.5%), followed by the lung (n = 11, 27.5%), peritoneum (n = 5, 12.5%), lymph nodes (n = 5, 12.5%), and bone (n = 1, 2.5%).

Baseline analytical parameters are summarized in Table 1. With reference to the hemogram, 30 patients (41%) presented with some degree of anemia and only 5 patients (6.8%) had thrombocytopenia. In relation to inflammatory parameters, 54 patients (74.0%) showed elevated CRP and the medians for NLR and PLR were 2.66 and 151.68, respectively. Most patients showed elevated levels of the tumor marker CA19-9 above the upper limit of normal (n = 49, 67.1%).

The molecular subtype was analyzed in 34 patients and the most common molecular subtype was classical (n = 17, 50%), followed by basal-like (n = 9, 26%), mixed (n = 5, 15%), and null (n = 3, 9%) subtypes.

### 2.2. Analysis of the Relationship between the Analyzed Cytokines and Other Analytical Parameters with Molecular Subtypes

Since we only had cytological samples available to analyze the molecular subtype of 34 patients, we grouped the classical subtype with the mixed subtype and the basal-like subtype with the null subtype to analyze the relationship of cytokines and other analytical parameters with molecular subtypes, as these molecular subtypes share certain characteristics and prognoses among themselves.

Elevated values of NLR, PLR, and TGF-β2 were associated with basal-like and null subtypes, with statistically significant differences (*p*-value = 0.003, *p*-value = 0.011 and *p*-value = 0.040, respectively), and elevated levels of IL-17A were associated with classical and mixed subtypes, with statistically significant differences (*p*-value = 0.021) (Figure 1). No differences were found between the rest of the cytokines and other analytical variables studied and the molecular subtype.

Subsequently, a multivariate analysis of molecular subtype was performed, in which it was observed that TGF-β2 levels were independently associated with basal-like and null subtypes. For each unit increase in TGFβ2, the risk of having basal-like or null subtypes increased by 0.3% (OR 1.003 [95% CI: 1.0001–1.006], *p*-value = 0.041). Regarding NLR and IL-17A, we cannot interpret the associated risks because they do not significantly affect the model.

With these results, we subsequently developed a predictive model for basal-like and null subtypes: log(p/1 − P) = −1.507 + 1.528 * NLR + 0.003 * TGFβ2 − 3.357 * IL17A. Here, P is the probability of presenting a basal-like or null subtype.

### 2.3. Survival Analysis of the Different Treatment Regimens Received and Relationship between the Molecular Subtype and Prognosis

Among the 73 patients included, 12 patients initially presented with localized/borderline disease and underwent surgery. Of these, 5 patients (41.7%) received neoadjuvant treatment, 3 patients (25%) received adjuvant treatment, 3 patients (25%) received both neoadjuvant and adjuvant treatment, and 1 patient (8.3%) did not receive any perioperative treatment. During follow up, 7 patients (58.3%) experienced disease recurrence and the median disease-free survival was 7.05 months (95% CI: 3.93–10.16).

In total, 68 out of 73 patients received first-line treatment, as 5 patients who underwent surgery did not experience disease recurrence. In relation to the first-line treatment, most patients received gemcitabine and nab-paclitaxel (n = 28, 41.2%), followed by FOLFIRINOX (n = 25, 36.8%), gemcitabine monotherapy (n = 11, 16.2%), and, less frequently, treatment within a clinical trial (n = 4, 5.9%). The best response was a partial response, achieved in one-third of patients (n = 23, 33.8%); stable disease was observed in 22 patients (32.4%), and there was disease progression in 23 patients (33.8%). In total, 63 patients (92.6%) progressed to first-line treatment, and the PFS was 6.10 months (95% CI: 4.08–8.11). In addition, 62 patients (84.9%) died, and the median OS was 12.53 months (95% CI: 9.18–15.89).

Regarding molecular subtype, patients with the classical subtype exhibited longer PFS (median of 8.67 months [95% CI: 2.72–14.61]) compared to those with the basal-like subtype (median of 2.67 months [95% CI: 1.88–3.45]), although statistically significant differences were not observed (*p*-value = 0.581) (Appendix A). Regarding OS, longer OS was observed in patients with the classical subtype (OS median of 24.34 months [95% CI: 10.13–38.59]) compared to patients with the basal-like subtype (OS median of 7.30 months [95% CI: 6.52–8.08]), although statistically significant differences were not observed either (*p*-value = 0.163) (Appendix A). On the other hand, the relationship between GATA6 H-score and CK17 H-score with survival was also analyzed, observing an association between CK17 H-score and OS. Specifically, for each unit increase in CK17 H-score, the risk of mortality increased by 0.8% (*p*-value = 0.048). No differences were found between GATA6 H-score and patient survival.

According to the molecular subtype and first-line chemotherapy regimen received, patients with the classical subtype who received first-line treatment with FOLFIRINOX showed longer PFS (median of 21.47 months [95% CI: 8.96–33.97]) than patients who received a gemcitabine-based regimen (median of 6.23 months [95% CI: 3.51–8.94]), but a *p*-value was not obtained due to the lack of cases and events. While patients with the basal-like subtype who received first-line treatment with a gemcitabine-based regimen had longer PFS times (median of 10.70 months [no confidence interval obtained]) than patients who received FOLFIRINOX (median of 2.40 months [95% CI: 2.36–2.43]), a *p*-value was not obtained due to a lack of cases and events.

### 2.4. Analysis of the Relationship between the Analyzed Cytokines and Survival

To analyze the relationship between cytokines and PFS during first-line treatment or OS, we only included the 61 patients with locally advanced and metastatic disease at diagnosis. We excluded patients with localized/borderline disease due to their different prognoses.

In the univariate analysis, increased levels of IL-1α, IL-1β, IL-6, IL-8, IL-9, and IL-15 were significantly associated with a higher risk of progression to first-line treatment. On the other hand, increased levels of IL-1β, IL-6, and IL-15 were associated with a higher risk of mortality, and increased levels of IL-8, IL-9 and TNF-α were also associated with a higher risk of mortality, but without statistically significant differences. Table 2 summarizes the risk of progression and death associated with each cytokine.

Subsequently, the percentiles (p) of cytokines associated with a higher risk of progression to first-line treatment were analyzed. Patients with IL-6 levels ≥ p75 showed a median PFS of 2.40 months, those with levels between p50 and p75 showed a median PFS of 2.90 months, those with levels between p25 and p50 showed a median PFS of 6.10 months, and patients with IL-6 levels < p25 showed a median PFS of 10.70 months, with statistically significant differences observed (*p*-value = 0.031) (Figure 2A). Having IL-6 levels between p50 and p75 increased the risk of progression by 2.23 times compared to values < p25 (HR 2.23 [95% CI: 1.03–4.79], *p*-value = 0.040). And having IL-6 levels ≥ p75 increased the risk of progression by 3.03 times compared to values < p25 (HR 3.03 [95% CI: 1.39–6.62], *p*-value = 0.005). Patients with IL-8 levels ≥ p75 had a median PFS of 1.87 months, those with IL-8 levels between p50 and p75 had a median PFS of 3.03 months, those with IL-8 levels between p25 and p50 had a median PFS of 9.30 months, and patients with levels < p25 had a median PFS of 10.70 months, with statistically significant differences observed (*p*-value < 0.001) (Figure 2B). Having IL-8 levels between p50 and p75 increased the risk of progression by 2.33 times compared to values < p25 (HR 2.33 [95% CI: 1.13–4.83], *p*-value = 0.022). And having IL-8 levels ≥ p75 increased the risk of progression by 4.61 times compared to values < p25 (HR 4.61 [95% CI: 2.04–10.40], *p*-value < 0.001). No differences were found in the analysis by percentiles of IL-1α, IL-1β, IL-9, and IL-15 and by PFS during first-line treatment.

In the same way, percentiles of cytokines associated with a higher risk of mortality were also analyzed. Patients with IL-8 levels ≥ p75 had a median OS of 5.90 months, those with levels between p50 and p75 had a median OS of 7.03 months, those with levels between p25 and p50 had a median OS of 14.63 months, and patients with IL-8 levels < p25 had a median OS of 15.83 months, with statistically significant differences (*p*-value = 0.002) (Figure 3). Having IL-8 levels ≥ p75 increased the risk of mortality by 3.79 times compared to values < p25 (HR 3.79 [95% CI: 1.68–8.56], *p*-value = 0.001). No differences were found in the analysis by percentiles of IL-1β, IL-6, IL-9, and IL-15; by TNF-α; and by OS.

Subsequently, a multivariate analysis was performed for PFS, considering the cytokines IL-6 and IL-8, and for OS, considering the cytokine IL-8. CA19-9 levels above the upper limit of normal were included in the multivariate analysis because there was an association between CA19-9 levels above the upper limit of normal and worse PFS during first-line treatment (HR 1.71 [95% CI: 0.97–2.99] *p* = 0.061) and OS (HR 2.17 [95% CI: 1.19–3.96] *p* = 0.011). Other clinical variables such as ECOG performance status, stage, and the chemotherapy regimen used during first-line treatment (FOLFIRINOX vs. gemcitabine-based regimen) did not show an association with PFS during first-line treatment or with OS, but they were included in the multivariate analysis due to their clinical relevance. Sex and age were not included in the multivariate analysis because they did not show an association with PFS during first-line treatment or with OS, and they were not considered clinically relevant.

The multivariate analysis showed that IL-8 levels and ECOG performance status were independently related to an increased risk of progression during first-line treatment (*p*-value < 0.001 and 0.038, respectively), and IL-8 levels were also independently associated with an increased risk of mortality (*p*-value = 0.007) (Table 3). Specifically, regarding PFS during first-line treatment, having IL-8 levels ≥ p75 increased the risk of progression by 7.92 times compared to values < p25 (HR 7.92 [95% CI: 3.22–19.48], *p*-value < 0.001), and having ECOG 1 or 2 increased the risk of progression by 1.97 times compared to ECOG 0 (HR 1.97 [95% CI: 1.03–3.74], *p*-value = 0.038). On the other hand, regarding OS, having IL-8 levels ≥ p75 increased the risk of mortality by 3.68 times compared to values < p25 (HR 3.68 [95% CI: 1.57–8.61], *p*-value = 0.003).

### 2.5. Analysis of the Relationship between the Analyzed Cytokines and Disease Stage at Diagnosis and Location of Metastases

Table 4 summarizes the concentration of each cytokine according to the disease stage at diagnosis. Elevated IL-8 levels were associated with the presence of metastatic disease at diagnosis (*p*-value = 0.008) (Figure 4). In the subgroup analysis, statistically significant differences were found in terms of IL-8 levels between patients with metastatic disease at diagnosis and those with localized/borderline disease (*p*-value = 0.024).

In relation to the localization of metastases, elevated levels of IL-6 and IL-8 were associated with the presence of liver metastases (*p*-value = 0.030 and *p*-value < 0.001, respectively, Figure 5). No significant differences were found between the rest of the cytokines studied and other metastatic locations.

## 3. Discussion

Many tumors originate in tissues where chronic inflammation exists. In addition, while in healthy individuals there is a fine balance between pro- and anti-inflammatory cytokines, contributing to homeostasis and wound healing, in cancer patients, during disease progression, this balance is disrupted, leading to a dysfunctional state of both immune stimulation and suppression [49,50]. This inflammatory response is regulated by a broad and complex network composed of cytokines that are altered in cancer patients [51]. Given the significant role that cytokines play in cancer development and progression, we decided to investigate their influence on the molecular subtype of patients with PDAC and their relationship with prognosis. The results obtained in our study highlight their association with molecular subtype and prognosis.

The most frequently occurring molecular subtype in our cohort was the classical subtype (50%), followed by basal-like (26.5%), mixed (14.7%), and null (8.8%) subtypes. These findings are consistent with previous research, which reported GATA6 expression (a surrogate marker of the classical subtype) in patients with advanced PDAC in 46% to 80% of cases and found this in patients with localized/borderline PDAC in 62% to 81% of cases [15,17,18,19,20,21,22,23,24]. The fact that, in our series, the proportion of patients with advanced PDAC was 83.6%, compared to the 16.4% of patients with localized/borderline PDAC, could explain the lower frequency of the classical subtype compared to other cohorts [15,17,18,19,20,21,22,23,24]. This variability in molecular subtypes also underscores the intrinsic heterogeneity of PDAC and the increasing importance of molecular characterization to guide treatment in these patients in the future [15,16,17,18,19,20,21,22,23,24]. Regarding the relationship between molecular subtype and prognosis in patients with PDAC, patients with a basal-like subtype showed a trend towards lower PFS and OS, but statistical significance was not reached, probably due to the small sample size of our cohort. The worse prognosis of basal-like patients has also been previously described in the literature [9,10,16,17,23,24]. Aung et al., in the COMPASS trial, described a median PFS of 2.3 months (95% CI, 1.8–6.0) for patients with the basal-like subtype, and a median OS of 6.3 months (95% CI, 4.0—not reached) [19]. Additionally, the null subtype has also been associated with poor prognosis and high metastatic potential in the literature [44]. In contrast, the classical subtype has been associated with a more favorable prognosis [21]. In the same way, we observed a significant association between CK17 H-score (expressed in the basal-like subtype) and OS, so that for each unit increase in CK17 H-score, the risk of death increased by 0.8%. This relationship between a higher CK17 H-score and worse prognosis in PDAC patients had not been previously described; thus, further studies are needed to assess its potential role as a prognostic biomarker in patients with PDAC. Furthermore, differences in patient survival were also found between classical and basal-like subtypes based on the first-line treatment regimen received. Patients with the classical subtype who received first-line treatment with FOLFIRINOX had a longer PFS than patients who received a gemcitabine-based regimen, but a *p*-value was not obtained due to a lack of cases and events. On the other hand, patients with a basal-like subtype who received first-line treatment with a gemcitabine-based regimen experienced longer PFS than patients who received FOLFIRINOX, but a *p*-value was not obtained either due to lack of cases and events. In the literature, patients with classical subtype also showed greater benefit from treatment with FOLFIRINOX, whereas patients with basal-like subtype achieved better results with gemcitabine-based regimens as well [9,10,16,17,23,24].

We additionally explored the relationship between cytokines and other analytical parameters with molecular subtypes of PDAC, and found that elevated values of NLR, PLR, and TGF-β2 were associated with basal-like and null subtypes, and elevated levels of IL-17A were associated with classical and mixed subtypes. In addition, a multivariate analysis revealed that TGF-β2 levels were independently associated with basal-like and null subtypes, allowing the development of a predictive model for basal-like and null subtypes. In terms of the literature, this is the first study that analyzes the relationship between cytokines and the molecular subtype. Furthermore, we found that elevated TGF-β2 levels were independently associated with basal-like and null subtypes, suggesting that TGF-β2 could be used as a biomarker for these molecular subtypes. However, these results need to be validated in future prospective studies with larger sample sizes. TGF-β is recognized as one of the main inducers of transcription factors involved in epithelial–mesenchymal transition, a distinctive feature of the basal-like subtype [52]. Therefore, there are previous data that partially support the relationship we observed in our study between TGF-β2 and basal-like and null subtypes [52]. These observations suggest that the involvement of cytokines in PDAC development may differ depending on molecular subtype. Furthermore, it could also be speculated that each molecular subtype and/or its stromal microenvironment may produce a different cytokine profile, contributing to tumor progression.

Another relevant finding of our study was the relationship between the analyzed cytokines and prognosis in patients with locally advanced and metastatic PDAC. We observed that the elevation of IL-1α, IL-1β, IL-6, IL-8, IL-9, and IL-15 was associated with a higher risk of progression during first-line treatment, and increased levels of IL-1β, IL-6, IL-8, IL-9, and IL-15 were related to increased mortality. Subsequently, in the percentile analysis, a statistically significant association was observed between higher percentiles of IL-6 and IL-8 and a shorter PFS during first-line treatment, and between higher percentiles of IL-8 and a shorter OS. Finally, in the multivariate analysis, only IL-8 levels were independently associated with the risk of progression during first-line treatment and the risk of death. These results support and, at the same time, add further relevance to the theory that cytokines provide information on the prognosis and evolution of patients with PDAC [30,31,32,33,34,35,36,37,38,39,40,41,42,43]. Specifically, in our study, the significant prognostic role of IL-8 must be highlighted, which has already been linked to a poorer prognosis in the literature [30,31,32,33,36]. IL-8 promotes angiogenesis and survival signals in cancer stem cells [36]. Additionally, it attracts suppressive cells derived from the myeloid lineage, contributing to the immunosuppressive microenvironment of PDAC [36]. It has also been associated with chemotherapy resistance, both to gemcitabine [53] and liposomal irinotecan [54]. Therefore, it is not surprising that, in our study, IL-8 has demonstrated its value in predicting time to progression and mortality, both in univariate and multivariate analyses. In addition, in the literature, other cytokines have also been associated with tumor progression and poorer prognosis in patients with PDAC [30,31,32,33,34,35,36,37,38,39,40,41,42,43]. Elevated IL-6 levels have been associated with tumor progression in various cancer types and with treatment response [31]. Additionally, IL-6 and IL-1β levels have been reported to relate to gemcitabine efficacy in PDAC patients [55]. These observations contrast with the results reported by Feng et al., who confirmed the prognostic value of IL-6 but not IL-1β [56]. In our study, we could verify that these two cytokines were related to both progression risk and mortality. Elevated TNF-α levels have been associated with chemoresistance and decreased survival in PDAC patients [33]. TNF-α is produced by tumor cells, stromal fibroblasts, and inflammatory cells. In addition, TNF-α stimulates the production of other cytokines and chemokines in the tumor microenvironment, which may contribute to tumor growth, angiogenesis, metastasis, and the development of chemoresistance [57]. In fact, it has been noted that transmembrane TNF-α expression was increased in PDAC cells treated with gemcitabine and paclitaxel compared to those not exposed to these drugs [58]. For all these reasons, these cytokines could have roles as biomarkers, stratifying progression risk, guiding therapeutic decisions, and serving as potential therapeutic targets for new treatments [30,31,32,33,34,35,36,37,38,39,40,41,42,43,44,45,46,47,48,53,54,55,56,57,58]. Similarly, the monitoring of these cytokines, especially IL-8, during the treatment of PDAC patients could provide information about treatment efficacy [30,31,32,33,36,53,54]. These results also support the idea that chronic inflammation plays an important role in the development and progression of PDAC [25,26,27,28,29]. Additionally, the inclusion of a percentile analysis of cytokine levels provides a novel approach and offers additional information on the relationship between cytokines and the prognosis of patients with PDAC, thus complementing the existing literature in this field.

Regarding the disease stage and the location of metastases, elevated levels of IL-6 and IL-8 were associated with the presence of liver metastases. Additionally, IL-8 was also associated with the presence of metastatic disease at diagnosis. On the other hand, there was no association between the rest of the analyzed cytokines and other metastatic locations. The literature has reported the association between certain cytokines and the development of metastases in patients with PDAC [37,38], but this is the first study to specifically reference the relationship between cytokines and the presence of liver metastases. IL-8 plays a significant role in angiogenesis, contributing to the formation of new blood vessels and facilitating tissue perfusion, which can promote the development of metastases in general, particularly liver metastases, the most frequent site of metastasis in PDAC [59]. These results may have clinical implications, as these cytokines can serve as potential biomarkers for the early detection of liver metastases in PDAC patients, and may also have therapeutic implications.

Some limitations of our study should be addressed. First, the small sample size is notable. Although a substantial number of patients were initially selected, unfortunately, 137 of them were excluded from the study due to various issues with the plasma samples. Specifically, the main reasons were that, for some patients, the samples were insufficient to perform cytokine analysis, and secondly, upon thawing some samples, we observed that they did not maintain the necessary integrity to conduct a rigorous cytokine study. This limitation could affect the generalization of our results and highlights the importance of sample collection, storage, and preservation procedures. Second, the use of multiplex technology instead of ELISA may have resulted in reduced sensitivity in detecting plasma levels of cytokines. On this point, another limitation would be that, despite using a broad panel of cytokines in our analysis, there are other cytokines that are also related to PDAC but which were not included, such as IL-4, IL-18, IL-20, IL-21, and IL-23, among others. Third, immunohistochemical analyses to determine the molecular subtype of PDAC patients were performed on cytological samples instead of tissue samples. The choice of this methodology was largely due to the challenges encountered in clinical practice in obtaining tissue biopsies from PDAC patients. Additionally, it is important to note that, due to these difficulties, we could only apply these immunohistochemical techniques to determine the molecular subtype in 34 patients. In addition, the evaluation and interpretation of IHC samples may be subject to some degree of subjectivity, which could pose an additional limitation in our study. Finally, its retrospective nature and the involvement of only one institution also represent limitations.

## 4. Materials and Methods

### 4.1. Patients

This is a single-institution retrospective observational study. We included all patients with PDAC from whom a plasma sample had been collected before receiving the first chemotherapy treatment at La Paz University Hospital between April 2016 and April 2022. The selection of patients whose plasma samples were deemed unsuitable was based on predefined criteria including insufficient sample volume, hemolysis, or contamination. We followed strict protocols to ensure the quality and integrity of the samples used in this study. Patients were aged 18 years or older and had an Eastern Cooperative Oncology Group (ECOG) performance status between 0 and 2. All living patients provided informed consent to the study, and, in addition, all patients had previously signed another informed consent authorizing the conservation of a plasma sample for research studies. The diagnosis of PDAC was confirmed cytologically.

This study was conducted in accordance with the ethical standards of the Helsinki Declaration by the World Medical Association and approved by the local Ethics Committee (code HULP: PI-4852).

### 4.2. Clinical and Analytical Variables Included

Data regarding clinical and demographic characteristics, tumor extension, location of metastases, and analytical parameters were obtained from the medical records of each patient. These measurements were taken during the first consultation of the medical oncology service before the initiation of the first cycle of chemotherapy. The median time from the first consultation to the first dose of chemotherapy was less than 15 days.

The following clinical variables of the patients were included: age, sex, ECOG performance status, medical history (history of smoking, diabetes mellitus, and chronic pancreatitis), details of diagnosis (stage and location of metastases), and treatments received (examining first-line treatment, type of response, and progression-free survival (PFS) during first-line treatment). The selection of the chemotherapy regimen for each patient was guided by clinical practice guidelines and also determined based on their ECOG performance status and comorbidities. Additionally, the date of death or last contact with the patient and the OS were collected. PFS after first-line treatment was evaluated from the date of the first cycle of treatment to the date of disease progression. OS was analyzed from the date of diagnosis to the date of death from any cause or last contact.

The subsequent analytical parameters were collected: hemoglobin, leukocytes (neutrophils and lymphocytes), platelets, C-reactive protein (CRP), and the tumor marker CA19-9. From these variables, the prognostic indexes of the neutrophil-to-lymphocyte ratio (NLR) and the platelet-to-lymphocyte ratio (PLR) were calculated. NLR was calculated by dividing the absolute neutrophil count by the absolute lymphocyte count and PLR was estimated by dividing the absolute platelet count by the absolute lymphocyte count. A high NLR and PLR are associated with worse prognosis and reduced survival in patients with PDAC [60,61]. The cut-off point used for NLR and PLR was the median.

### 4.3. Processing of Cytology Samples

The expression of GATA6 and CK17 was determined by immunohistochemistry (IHC) in cytological samples, collected at the time of diagnosis and archived in the Pathology Department.

For the inclusion of the cells in paraffin, the cells were first collected in Thinprep™Cytolyt^®^ medium (Hologic, Madrid, Spain), a methanol-based buffered solution for cell washing that lyses red blood cells, prevents protein precipitation, and dissolves mucus to preserve cell morphology before processing. Next, the cells were centrifuged and fixed in 96% ethanol for 2–3 h, after which they were centrifuged again and placed in a cassette that was introduced into a processor for paraffin embedding and subsequent block generation.

Cytological blocks were evaluated and selected by an expert pathologist. Subsequently, 4 µm sections were cut, which were deparaffinized in xylene and rehydrated in a series of decreasing ethanol concentrations (100–70%). Antigen retrieval was then performed using the PT Link system (Agilent, Santa Clara, CA, USA). After this step involving antigen, endogenous peroxidase was blocked for 10 min, and samples were incubated with primary antibodies (GATA6, #AF1700 R&D Systems (Minneapolis, MN, USA) and CK17 (EP1623), #ab109725, Abcam (Cambridge, UK)) and secondary antibodies (Anti-Goat, #P044901-2 and Dual rabbit/mouse HRP, # K5007, both from Agilent). Lastly, sections were visualized by applying the diaminobenzidine (DAB) chromogen and performing counterstaining with hematoxylin. All compounds used for IHC were from the Agilent EnVision FLEX kit (Agilent). Images were acquired with an Olympus BX51 microscope, coupled with an Olympus DP70 digital camera (Olympus, Shinjuku, Tokyo, Japan).

Evaluation was performed by a specialist pathologist, considering the staining intensity and the percentage of stained tumor cells. For GATA6, nuclear staining was evaluated, while for CK17, membrane staining was evaluated. Intensities were categorized as follows: 0 (negative), 1 (weak), 2 (intermediate), and 3 (strong). The H-score was calculated for both biomarkers by multiplying the staining intensity by the percentage of positive cells. The H-score value ranged from 0 to 300. An H-score < 30 was considered negative [62,63].

Subsequently, considering the H-score of GATA6 and CK17, 4 molecular subtypes were established: classical (GATA6 H-score ≥ 30 and CK17 H-score < 30), basal-like (CK17 H-score ≥ 30 and GATA6 H-score < 30), mixed (GATA6 and CK17 H-score ≥ 30), and null (GATA6 and CK17 H-score < 30) [44]. Appendix A show examples of immunohistochemistry for GATA6 and CK17 for classical, basal-like, and mixed subtypes.

### 4.4. Processing of Plasma Samples and Cytokine Analysis

In the Clinical Analysis Department of our hospital, the plasma levels of the following cytokines were measured: transforming growth factor beta (TGF-β)1-3, IL-1α, IL-1β, IL-6, IL-8, IL-9, IL-10, interferon-induced protein 10 (IP-10), IL-15, IL-17A, and TNF-α. A total of 500 milliliters (mL) of plasma from each patient was used. The analysis of the cytokines was performed using the Millipore-Luminex^®^ 200™ analyzer. This platform uses a multiplex bead-based immunoassay. The kit used was HCYTOMAG-60K (Merck Millipore, Burlington, MA, USA). This kit includes magnetic microspheres, primary antibodies that specifically bind to the target cytokines, and secondary antibodies conjugated with fluorophores for cytokine detection.

In the initial step, magnetic microspheres were coated with primary antibodies specific to the target cytokines. Subsequently, antibody-coated microspheres were mixed with the sample containing the cytokines. During the first incubation, cytokines were bound to the antibodies on the microspheres. After the first incubation, a wash step was conducted. Then, we carried out a second incubation with secondary antibodies labeled with fluorophores, which bound to the cytokines captured on the microspheres, allowing the subsequent detection and quantification of the cytokines. After the second incubation, another wash step was performed. Finally, the microspheres were analyzed on the Luminex^®^ 200™ system, which detected and quantified the fluorescence of each microsphere, providing information about the number of cytokines present in each sample.

### 4.5. Statistical Analysis

The description of qualitative data was conducted in the form of absolute frequencies and percentages, and quantitative data were described using mean and standard deviation or median and interquartile range, depending on the distribution of the data. The normality of continuous variables was assessed using the Kolmogorov–Smirnov test.

For the comparison of qualitative variables, Pearson’s Chi-Square test was utilized. The association between qualitative and quantitative variables was examined using the Student’s *t* test or one-way ANOVA (depending on whether two or more groups were compared), or their non-parametric equivalents (Mann–Whitney U test or Kruskal–Wallis test), with Bonferroni correction applied for multiple comparisons. To explore the relationship between quantitative variables, Pearson correlation or its non-parametric counterpart, Spearman correlation, was conducted.

Survival was estimated using the Kaplan–Meier method and described using median values with 95% confidence intervals (CIs). A Cox regression was performed to estimate the hazard ratios (HR) and the 95% CI.

Multivariate binary logistic regression analysis was applied to obtain a predictive model for the molecular subtype variable. This model was adjusted for variables that were individually significant. The optimal model was selected using the AIC criterion.

All the tests were two-sided, and *p*-values < 0.05 were considered statistically significant. Data analysis was performed using the statistical software R version 4.3.3 (R Core Team, 2020).

## 5. Conclusions

The results of our study suggest that cytokine expression differs according to the molecular subtype of PDAC, with the basal-like subtype being characterized by an association with an inflammatory context, showing high levels of TGF-β2 and other markers of systemic inflammation, such as NLR and PLR. On the other hand, elevated levels of IL-17A were associated with the classical subtype. Furthermore, determining the levels of TGF-β2, IL-17A, and NLR may help to identify the basal-like subtype through a predictive model. Additionally, cytokines play a relevant role in the development of metastases and the prognosis of patients with PDAC. However, due to the small sample size upon which these statements are based, larger studies are needed to confirm these findings.

## Figures and Tables

**Figure 1 ijms-25-09368-f001:**
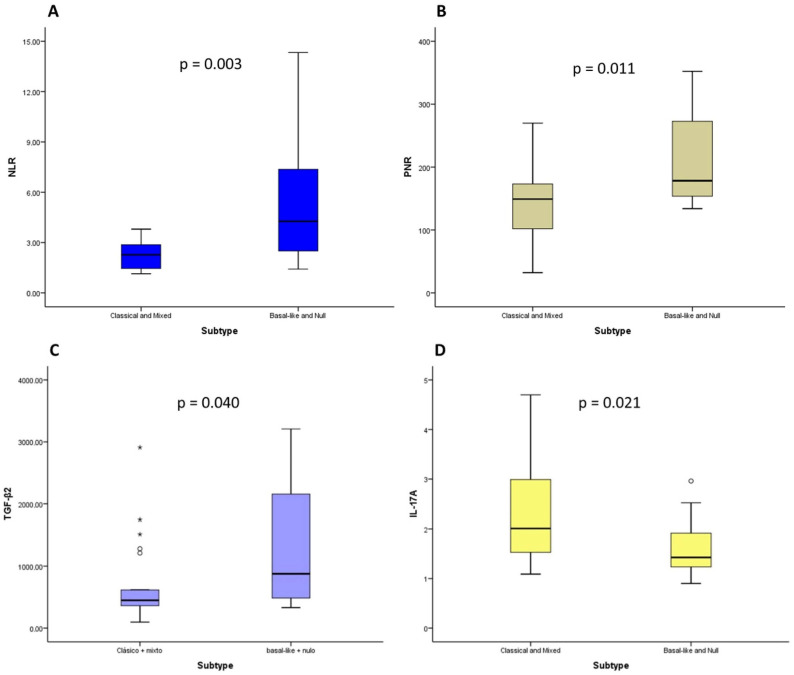
Relationship between values of neutrophil-to-lymphocyte ratio (NLR) (**A**), platelet-to-lymphocyte ratio (PLR) (**B**), TGF-β2 (**C**), and IL-17A (**D**) and molecular subtypes in patients with pancreatic ductal adenocarcinoma (PDAC). The asterisks “*” are outliers, values that deviate from the rest of the values taken by the variable.

**Figure 2 ijms-25-09368-f002:**
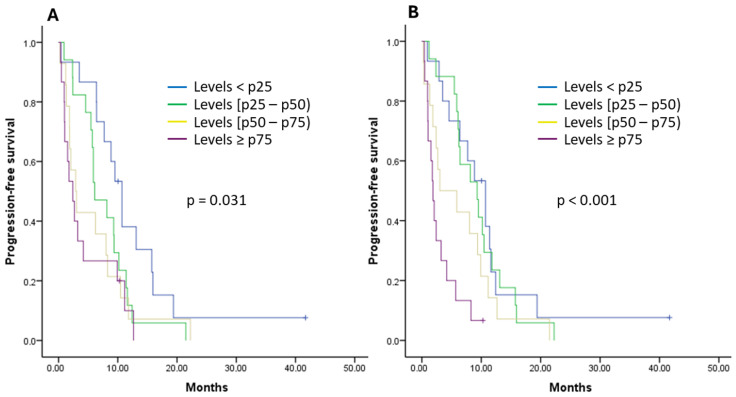
Kaplan–Meier curves for progression-free survival (PFS) during first-line treatment in patients with pancreatic ductal adenocarcinoma (PDAC) based on cytokine percentiles: IL-6 (**A**) and IL-8 (**B**).

**Figure 3 ijms-25-09368-f003:**
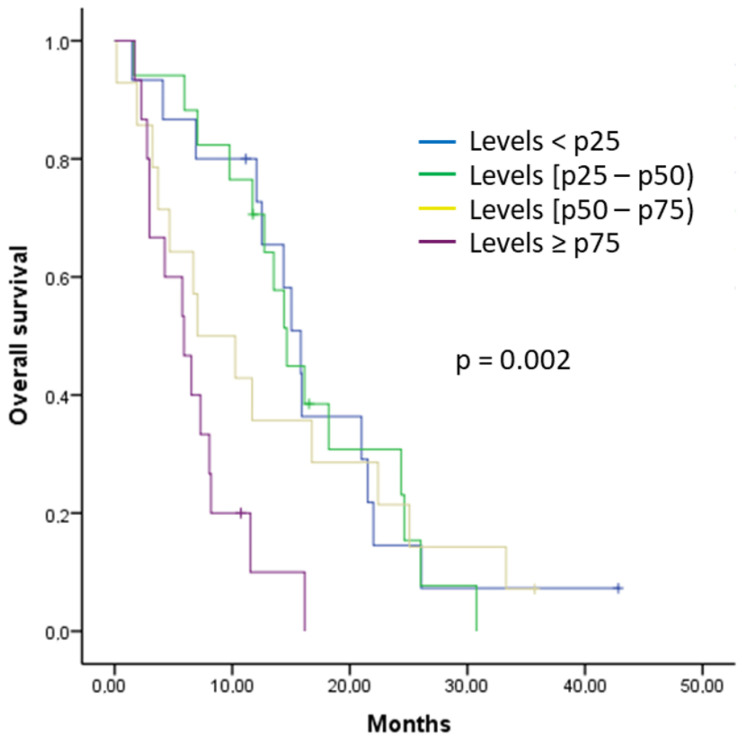
Kaplan–Meier curve for overall survival (OS) in patients with pancreatic ductal adenocarcinoma (PDAC) based on percentiles of IL-8.

**Figure 4 ijms-25-09368-f004:**
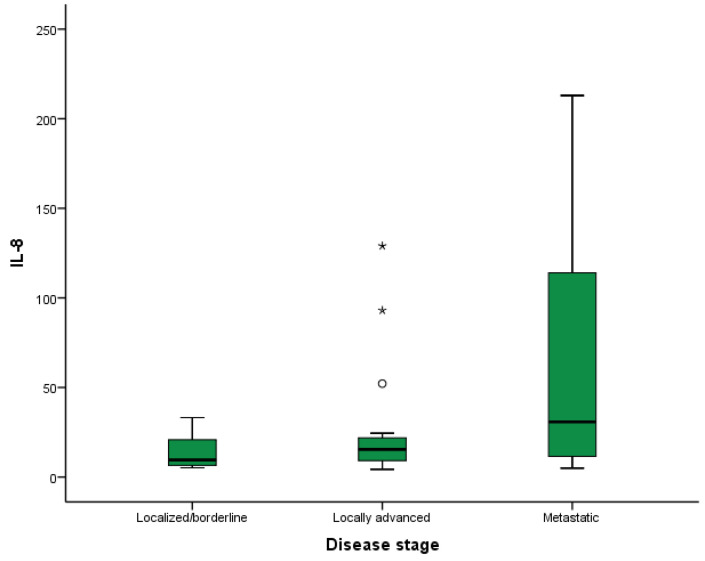
Relationship between IL-8 levels and disease stage in patients with pancreatic ductal adenocarcinoma (PDAC). The asterisks “*” are outliers, values that deviate from the rest of the values taken by the variable.

**Figure 5 ijms-25-09368-f005:**
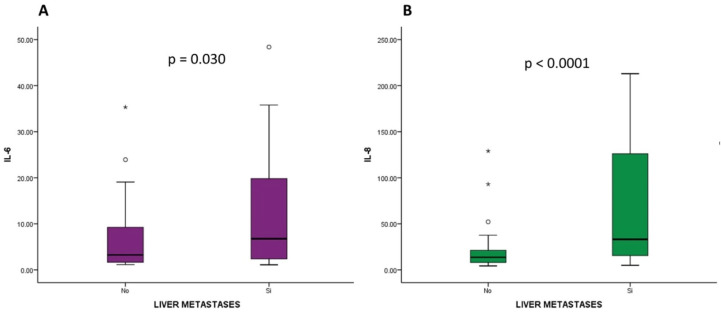
Relationship between IL-6 levels (**A**) and IL-8 levels (**B**) and presence of liver metastases in patients with pancreatic ductal adenocarcinoma (PDAC). The asterisks “*” are outliers, values that deviate from the rest of the values taken by the variable.

**Table 1 ijms-25-09368-t001:** Baseline characteristics, clinical stage, analytical parameters, and molecular subtype classification of patients.

Variables	Patients (%)
Sex	n 73
Male	33 (45)
Female	40 (55)
ECOG performance status at diagnosis	
0	24 (33)
1	45 (62)
2	4 (5)
Smoking history	
Smoker	21 (29)
Former smoker	25 (34)
Never smoker	27 (37)
Diabetes mellitus	26 (36)
Chronic pancreatitis	6 (8)
Primary tumor location	
Head	41 (56)
Body and tail	32 (44)
Stage	
Localized/borderline	12 (16)
Locally advanced	21 (29)
Metastatic	40 (55)
Hemoglobin	
Normal hemoglobin (≥12.5 in males and ≥11.8 in females)	43 (58.9)
Anemia grade 1 (Normal hemoglobin—Hb 10 gr/dL)	28 (38.3)
Anemia grade 2 (Hb 9.9 gr/dL—Hb 8 gr/dL)	1 (1.4)
Anemia grade 3 (Hb 7.9 gr/dL—Hb 6.5 gr/dL)	1 (1.4)
Platelets	
<150.000/µL	5 (6.8)
≥150.000/µL	68 (93.2)
C-reactive protein (CRP)	
>5 mg/L	54 (74.0)
≤5 mg/L	19 (26.0)
Neutrophil-to-lymphocyte ratio (NLR)	
>Median	37 (50.7)
≤Median	36 (49.3)
Platelet-to-lymphocyte ratio (PLR)	
>Median	36 (49.3)
≤Median	37 (50.7)
CA19-9	
>36 UI/mL	49 (67.1)
≤36 UI/mL	24 (32.9)
Molecular subtype	n 34
Classical	17 (50)
Basal-like	9 (26)
Mixed	5 (15)
Null	3 (9)

**Table 2 ijms-25-09368-t002:** Univariate analysis of progression-free survival (PFS) during first-line treatment and overall survival (OS) considering the analyzed cytokines.

Cytokines	Univariate Analysis for PFS during First-Line		
	*p*-Value	HR	95% CI
IL-1α	**0.011**	**1.004**	1.001–1.006
IL-1β	**0.001**	**1.439**	1.150–1.801
IL-6	**<0.001**	**1.016**	1.008–1.023
IL-8	**0.045**	**1.001**	1.00003–1.003
IL-9	**0.004**	**1.010**	1.003–1.017
IL-10	0.880	0.997	0.954–1.041
IP-10	0.169	1.0001	0.999–1.0004
IL-15	**<0.001**	**1.210**	1.110–1.318
IL-17A	0.377	1.093	0.897–1.333
TGF-β1	0.305	1.00001	0.999–1.00004
TGF-β2	0.719	1.00005	0.999–1.0003
TGF-β3	0.486	0.998	0.993–1.003
TNF-α	0.192	1.007	0.996–1.018
**Cytokines**	**Univariate Analysis for OS**		
	***p*-Value**	**HR**	**95% CI**
IL-1α	0.096	1.002	0.999–1.005
IL-1β	**0.023**	**1.298**	1.036–1.626
IL-6	**0.001**	**1.012**	1.005–1.019
IL-8	**0.058**	**1.001**	0.999–1.003
IL-9	**0.054**	**1.007**	0.999–1.014
IL-10	0.717	1.008	0.967–1.050
IP-10	0.231	1.0001	0.999–1.0003
IL-15	**<0.001**	**1.189**	1.086–1.302
IL-17A	0.478	1.075	0.881–1.311
TGF-β1	0.146	1.00002	0.999–1.00005
TGF-β2	0.730	1.00005	0.999–1.0003
TGF-β3	0.320	0.997	0.992–1.003
TNF-α	**0.052**	**1.012**	0.999–1.023

Significant results are in bold.

**Table 3 ijms-25-09368-t003:** Multivariate analysis of progression-free survival (PFS) during first-line treatment, considering IL-6 and IL-8, and overall survival (OS), considering IL-8.

Multivariate Analysis for PFS during First-Line			
Variables	*p*-Value	HR	95% CI
**Initial model**			
ECOG performance status	0.200	1.62	0.77–3.41
Stage	0.786	1.09	0.57–2.07
Chemotherapy regimen used during first-line treatment *	0.406	1.31	0.68–2.51
CA19-9 levels above the upper limit of normal	0.180	1.55	0.81–2.98
IL-6 (reference group < p25)	0.676		
− [p25–p50)	0.276	1.65	0.67–4.07
− [p50–p75)	0.289	1.67	0.64–4.35
− ≥p75	0.322	1.81	0.55–5.92
IL-8 (reference group < p25)	0.025		
− [p25–p50)	0.967	1.01	0.45–2.29
− [p50–p75)	0.431	1.52	0.53–4.33
− ≥p75	0.009	4.95	1.48–16.54
**Final model**			
ECOG performance status	**0.038**	**1.97**	1.03–3.74
IL-8 (reference group < p25)	**<0.001**		
− [p25–p50)	0.848	1.07	0.51–2.23
− [p50–p75)	0.130	1.80	0.84–3.85
− ≥p75	**<0.001**	**7.92**	3.22–19.48
**Multivariate analysis for OS**			
**Variables**	***p*-Value**	**HR**	**95% CI**
**Initial model**			
ECOG performance status	0.082	1.82	0.92–3.58
Stage	0.721	0.89	0.48–1.65
Chemotherapy regimen used during first-line treatment *	0.955	0.98	0.53–1.79
CA19-9 levels above the upper limit of normal	0.071	1.83	0.94–3.56
IL-8 (reference group < p25)	0.008		
− [p25-p50)	0.901	0.95	0.44–2.04
− [p50–p75)	0.663	1.20	0.52–2.76
− ≥p75	0.003	3.84	1.58–9.34
**Final model**			
ECOG performance status	0.080	1.78	0.93–3.41
CA19-9 levels above the upper limit of normal	0.064	1.86	0.96–3.59
IL-8 (reference group < p25)	0.007		
− [p25-p50)	0.872	0.93	0.43–2.01
− [p50–p75)	0.721	1.15	0.51–2.59
− ≥p75	**0.003**	**3.68**	1.57–8.61

Significant results are in bold. * FOLFIRINOX vs. gemcitabine-based regimen.

**Table 4 ijms-25-09368-t004:** Percentile analysis of cytokine concentration by stage at diagnosis and univariate analysis of relationship between each cytokine and stage at diagnosis.

Cytokines	Localized/Borderline Disease			Locally Advanced Disease			Metastatic Disease			*p*-Value
	p25	p50	p75	p25	p50	p75	p25	p50	p75	
IL-1α	0.80	1.67	3.15	0.45	0.76	1.03	0.58	1.00	3.43	0.128
IL-1β	1.25	1.79	2.25	1.21	1.51	1.99	1.12	1.37	1.80	0.481
IL-6	1.35	2.24	4.56	1.95	5.48	11.30	2.29	6.29	20.20	0.112
IL-8	6.45	9.58	20.97	8.91	15.40	22.92	11.41	30.81	117	0.008
IL-9	1.35	3.45	6.98	1.73	2.73	6.15	1.53	2.03	3.18	0.557
IL-10	3.50	5.56	6.76	3.33	4.27	5.41	2.74	3.86	6.85	0.737
IP-10	369.00	437.50	958.25	444	637	1502	415	661	1087.25	0.444
IL-15	3.16	4.95	5.75	2.73	3.62	4.59	2.90	4.09	5.45	0.432
IL-17A	1.42	2.16	2.89	1.55	2.17	2.90	1.33	1.55	2.02	0.057
TGF-β1	8102.75	12,553.50	32,262	6581.50	15,285	21,542	8243	14,496.50	21,854	0.791
TGF-β2	409.00	566.00	1629.75	330	559	1573.50	428	639	1099.25	0.944
TGF-β3	30.92	39.94	87.89	36.62	72.36	132	33.30	63.71	72.36	0.417
TNF-α	8.67	12.09	25.18	8.16	17.69	23.52	10.70	16.31	30.21	0.475

p: percentile.

## Data Availability

The data that support the findings of this study are available on request from the corresponding author. The data are not publicly available due to privacy or ethical restrictions.

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
