# Peer review of "Cytokines and Pancreatic Ductal Adenocarcinoma: Exploring Their Relationship with Molecular Subtypes and Prognosis"

_ijms, 2024, doi:10.3390/ijms25179368_

Round 1

Reviewer 1 Report

Comments and Suggestions for Authors

The manuscript titled ‘Cytokines and Pancreatic Ductal Adenocarcinoma: Exploring Their Relationship with Molecular Subtypes and Prognosis” by Gutierrez-Sainz et al.,  is a descriptive study that shows IL-8 and IL-6  are associated with the basal- and null-type PDA and poor prognosis whereas TGF-β2 and IL-17A were associated with classical and mixed subtypes. Other cytokines were less clearly associated. The data are suggestive and encourage more patient studies to allow firm conclusions to be made about their value as prognostic markers.  Biological mechanisms influencing these associations were not investigated.

Minor comment:

1.     Define the Null subtype. How does it differ from Classical and Basal-like subtypes?  

2.     What does “mixed” refer to? – tumors with Classical features transitioning to Basal, or Null, or both?

Author Response

To Whom It May Concern,

Thank you for your valuable reviews and questions. I appreciate the opportunity to clarify these points.

  1. Define the Null subtype. How does it differ from Classical and Basal-like subtypes?

I have added a definition of the null subtype in the introduction. The null subtype is characterized by the lack of expression of GATA6, CK5, and CK17. This distinguishes it from the classical and basal-like subtypes, where classical typically expresses GATA6 but not CK5/CK17, and Basal-like expresses CK5/CK17 but not GATA6.

  1. What does “mixed” refer to? – tumors with Classical features transitioning to Basal, or Null, or both?

I have also added a definition of the mixed subtype in the introduction. The "mixed" subtype refers to tumors that express both GATA6 and CK5/17. This means they exhibit features of both Classical and Basal-like subtypes, rather than transitioning specifically to Null.

Thank you again for your insightful questions.

Best regards,

Laura Gutiérrez

Reviewer 2 Report

Comments and Suggestions for Authors

I have been asked to review the manuscript titled “Cytokines and Pancreatic Ductal Adenocarcinoma: Exploring Their Relationship with Molecular Subtypes and Prognosis” by Laura Gutierrez-Sainz et al. The authors aimed to evaluate the association between cytokines and molecular subtypes of PDAC and determine their prognostic value, a significant area of interest in the fight against pancreatic cancer.

I have decided not to recommend this manuscript for publication based on several critical issues.

1. The study sample size is very small, particularly for the analysis of molecular subtypes (34 patients), which affects the statistical power. Moreover, the high attrition rate from the initial 210 patients to 73 usable samples (with only 34 having molecular subtype data) raises concerns about selection bias, which was not adequately addressed.

2. The retrospective observational design and single-institution setting limit the ability to establish causality between cytokine levels and PDAC molecular subtypes or prognosis. The findings may not be generalizable to other populations or clinical settings.

3. The methods section lacks clarity on the criteria used for selecting patients whose plasma samples were deemed unsuitable. Furthermore, the descriptions of the immunohistochemistry procedures and the handling of cytological samples could be more detailed.

4. The study does not convincingly establish the clinical utility of measuring cytokine levels for prognostic purposes in PDAC.

5. Figures and tables are not well-integrated into the text, and their descriptions are often insufficient. The information on the pictures are not also sufficient.

Comments on the Quality of English Language

The manuscript has several typographical and grammatical errors. 

Author Response

To Whom It May Concern,

Thank you for your thoughtful feedback. I appreciate the opportunity to clarify these points.

  1. The study sample size is very small, particularly for the analysis of molecular subtypes (34 patients), which affects the statistical power. Moreover, the high attrition rate from the initial 210 patients to 73 usable samples (with only 34 having molecular subtype data) raises concerns about selection bias, which was not adequately addressed.

We acknowledge the concerns regarding the small sample size and the potential for selection bias. Although a substantial number of patients were initially selected, unfortunately, 137 of them were excluded from the study due to various issues with the plasma samples. Specifically, the main reasons were that, for some patients, the samples were insufficient to perform cytokine analysis, and secondly, upon thawing some samples, we observed that they did not maintain the necessary integrity to conduct a rigorous cytokine study.

This limitation could indeed affect the generalization of our results and highlights the importance of sample collection, storage, and preservation procedures. We recognize that the high attrition rate and the resultant small sample size for the analysis of molecular subtypes (34 patients) impact the statistical power and raise concerns about selection bias. We have now addressed this potential bias in the revised manuscript by discussing the reasons for sample exclusion and the implications for our findings in the discussion section.

  1. The retrospective observational design and single-institution setting limit the ability to establish causality between cytokine levels and PDAC molecular subtypes or prognosis. The findings may not be generalizable to other populations or clinical settings.

We acknowledge that the retrospective nature of this study and its execution within a single institution present inherent limitations. These factors may affect the potential reproducibility of our results in other populations. Our findings could be influenced by characteristics specific to our institution, such as geographical location, available resources, clinical practices, and the patient population served. Additionally, due to the retrospective design, we cannot rule out the influence of unmeasured confounding factors, such as dose reductions or treatment interruptions at the discretion of the treating physician, which cannot be adequately controlled outside of a clinical trial.

Despite these limitations, we believe that our results provide a valuable foundation for the development of larger, prospective trials to confirm these findings.

Thank you for your understanding. We hope these clarifications address your concerns, and we appreciate your consideration of our manuscript.

  1. The methods section lacks clarity on the criteria used for selecting patients whose plasma samples were deemed unsuitable. Furthermore, the descriptions of the immunohistochemistry procedures and the handling of cytological samples could be more detailed.

In response to your feedback regarding the methods section, I would like to provide the following clarifications and justifications:

The selection of patients whose plasma samples were deemed unsuitable was based on predefined criteria including insufficient sample volume, hemolysis, or contamination. We followed strict protocols to ensure the quality and integrity of the samples used in the study. I have expanded this information in the methods section.

Regarding the immunohistochemistry procedures and the handling of cytological samples, we have now included more detailed descriptions in the revised manuscript. These procedures adhered to standardized protocols to ensure consistency and reliability of the results.

  1. The study does not convincingly establish the clinical utility of measuring cytokine levels for prognostic purposes in PDAC.

Thank you for your feedback. In response to your concern, I have expanded the information regarding the prognostic role of cytokines in PDAC in the introduction section. I hope this additional context better supports the clinical utility of measuring cytokine levels for prognostic purposes in this disease.

  1. Figures and tables are not well-integrated into the text, and their descriptions are often insufficient. The information on the pictures are not also sufficient.

Thank you for your valuable feedback. I have reviewed and revised the titles and descriptions of all tables and figures to ensure they are better integrated into the text and provide sufficient information. The revised titles now include more detailed descriptions to enhance clarity and context for each visual element.

I appreciate your comments, which have been instrumental in improving the quality of the manuscript.

Thank you again for your valuable comments.

Best regards,

Laura Gutiérrez

Reviewer 3 Report

Comments and Suggestions for Authors

This manuscript aims to study the relationship between cytokines and the molecular subtype of pancreatic ductal adenocarcinoma (PDAC). To reach this aim, the authors analyzed cytokine levels in 73 patients and molecular subtypes of 34 patients. Although interesting, this study does not have a good scientific soundness. First, cytokines are crucial in the pathogenesis of many diseases, such as cancer and inflammation. Using cytokine levels as PDAC prognosis is not disease specific with low accuracy. Second, although researchers found that there is an association between elevated levels of some cytokines. I do not see any experimental validation, so the observation can only be treated as a scientific hypothesis. Taken together, this study presents an interesting retrospective analysis. However, the whole study lacks specificity and accuracy, as many diseases have elevated cytokines, and there is no experimental support.      

Comments on the Quality of English Language

English is fine. 

Author Response

To Whom It May Concern,

We sincerely appreciate your thoughtful comments and the time you have taken to review our manuscript. Your observations regarding the specificity and accuracy of our study, as well as the need for experimental validation, are highly valued.

Firstly, we acknowledge that cytokines play a crucial role in the pathogenesis of many diseases, including cancer and inflammatory conditions. However, this very characteristic underscores the relevance of our study. The identification of specific cytokine profiles associated with molecular subtypes of PDAC could provide valuable insights to enhance the precision of prognostic assessments in this disease, which, as you know, has one of the lowest survival rates among all cancers. While elevated cytokine levels are not exclusive to PDAC, our focus is on the association of these cytokines with specific molecular subtypes within PDAC, which could help identify distinct patterns within this pathology.

Secondly, regarding the lack of experimental validation, we would like to emphasize that our study is based on a comprehensive clinical analysis of a cohort of PDAC patients, representing a crucial first step in identifying biomarkers with potential prognostic value. Retrospective studies like ours are fundamental in generating hypotheses that can later be validated through larger experimental and clinical studies. In this context, our research not only provides relevant data but also lays a solid foundation for future experimental studies that can validate and expand upon our findings.

In light of your feedback, we have made several modifications to the manuscript to improve clarity and address these concerns. These changes aim to better highlight the study’s significance and the potential implications of our findings.

Thank you again for your valuable comments.

Best regards,

Laura Gutiérrez

Round 2

Reviewer 2 Report

Comments and Suggestions for Authors

Dear Laura Gutiérrez,

Thank you for your detailed response and the revisions you have made to the manuscript. I have reviewed your clarifications and updates, and they are acceptable. 

Best regards,